# GOAL2FLOWNET: LEARNING DIVERSE POLICY COVERS USING GFLOWNETS FOR GOAL-CONDITIONED RL

## ABSTRACT

Goal-Conditioned Reinforcement Learning is a promising direction for learning policies to reach a diverse set of goals and achieve a flexible and adaptable agent capable of solving multiple tasks. However, many current approaches train policies that explore only a subset of the state space or learn to achieve only a subset of goals in a limited number of ways, leading to suboptimality in these learned policies. This leads to brittleness when the agent is taken into new regions of the state space, or when the distribution changes, rendering the learnt policy ineffective. Additionally, we argue that this also leads to poor sample efficiency and convergence because the knowledge for a specific set of goals has worse generalization to other goals. We propose *Goal2FlowNets*, that use Generative Flow Networks (GFlowNets) in order to learn exploratory goal-conditioned policies that are robust and can generalize better by learning multiple nearly optimal paths to reach the goals. We show that this leads to a significant improvement in sample complexity and enables better zero-shot and few-shot generalization to novel environmental changes through the learning of a stochastic goal-conditioned policy that has a wide coverage of the state and goal space.

## 1 INTRODUCTION

A remarkable characteristic of human intelligence is the ability to learn to do a variety of tasks with a limited amount of experience. Moreover, the knowledge to do one set of tasks can be used to learn other similar, but novel, tasks in an efficient manner. Goal-conditioned Reinforcement Learning is one promising direction in which an agent can be trained to achieve multiple tasks and reach a number of goals from a given goal distribution. Each of these goals can be thought of as a separate task, and thus goal-conditioned policies naturally learn to solve a multi-task problem where a goal and its corresponding reward define a task. Since goals typically share structure with their environment, it is natural for generalizations to occur. However, even with shared properties across goals, a large number of current goal-conditioned reinforcement learning methods learn policies that achieve a given goal in a limited number of ways, even when multiple ways exist. An example illustrating this problem is shown in Fig. 1.

Different techniques have been used to encourage exploration in reinforcement learning, such as rewarding novel states through random network distillation (Burda et al., 2018; Bellemare et al., 2016), maximizing state entropy (Liu & Abbeel, 2021), maximizing a learned prediction error (Pathak et al., 2017), or a combination of the above (Badia et al., 2020; Zhang et al., 2021; Wan et al., 2023). These exploration-based methods are promising but have been shown to learn fairly low-entropy "single-minded" policies at convergence, even when multiple optimal ways exist to complete a given task. This is due to optimizing the extrinsic reward for the underlying RL policy, which eventually only optimizes for one way of completing the task.

Planning methods are efficient, but tend to suffer when presented with changes to the environment dynamics. A method that is exploratory in nature and can leverage the structure in the underlying distribution to produce a diversity of paths to a goal can be advantageous to not only achieve a more strategic exploration, but also do so in a sample efficient and robust manner. An agent that knows multiple ways to accomplish a goal will still be able to function if some of those ways become invalid. Such "diversity-minded" agents may also improve generalization through the simple exposure to a broader distribution of intermediate states.

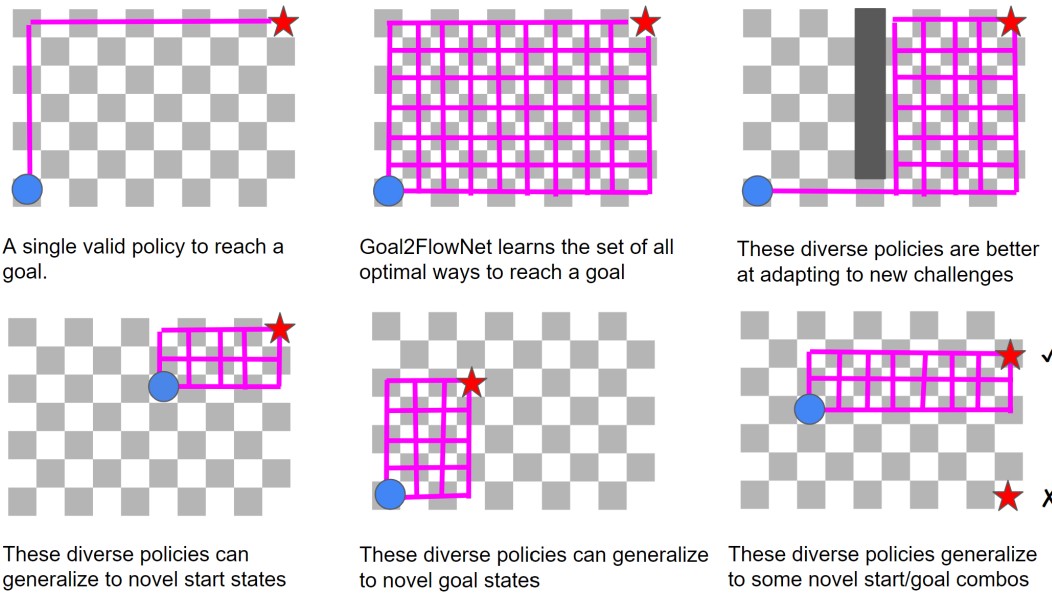

Figure 1: Goal2FlowNets learns a diverse set of policies from various start states (blue circles) to goal states (red stars).

In this work, we introduce **Goal2FlowNets**. Generative Flow Networks (GFlowNets) (Bengio et al., 2021; 2023) were introduced as generative methods that can sample a diverse set of trajectories from a given energy function or reward function. Goal2FlowNets uses GFlowNets to learn a goal-conditioned GFlowNet policy that generalizes over goals, and without preference over trajectories that achieve such goals. We show that due to the off-policy learning capabilities of GFlowNets, which also implicitly capture epistemic uncertainty, a Goal2FlowNets policy achieves better sample efficiency while still exploring the underlying state and goal distribution, and achieves better generalization to out-of-distribution scenarios over a large number of sparse-reward partial-observation based environments from the MiniGrid and BabyAI suites.

Our proposed method enables better exploration and learns multiple ways to complete goals drawn from a goal distribution. This property of our method results in attaining better generalization to novel configurations not seen during training, both zero-shot and few-shot, to enable better robustness and quick adaptation to novel changes to the distribution.

## 2 RELATED WORK

**GFlowNets:** GFlowNets have been proven to be versatile and effective in a number of applications, such as molecule discovery (Bengio et al., 2021), causal discovery (Deleu et al., 2022), multi-objective optimization (Jain et al., 2023), and combinatorial optimization (Zhang et al., 2023). There are many objectives to train GFlowNets, flow matching (Bengio et al., 2021), detailed balance (DB, Bengio et al., 2023), and trajectory balance (TB, Malkin et al., 2022), which optimizes GFlowNets at the trajectory level, but induces larger variance () even in deterministic MDPs; the problem is further exacerbated in stochastic tasks (Pan et al., 2023). Sub-trajectory balance (Madan et al., 2023) tackles the high variance problem at a sub-trajectory level and results in lower variance and higher sample efficiency. We use this last objective in our work.

Prior works have successfully used conditioning in GFlowNets to make them learn a family of distributions. Multi-Objective GFNs (Jain et al., 2023) are conditioned on a vector of preferences that scalarizes a vector reward. Roy et al. (2023) refine this approach by conditioning the model on a region of this *reward space*, giving it 0 reward otherwise. These are in some sense goals, but in (multiobjective) reward space rather than state space; our work specifically deals with RL-type state-related goals. Hu et al. (2023) train conditional models in a discrete latent variable autoencoder setting. GFlowOut (Liu et al., 2023) predicts dropout masks conditioned on the activations of a layer. As far as we know, we are the first to use GFlowNets as policies in a typical RL setting, and one

more complex than gridworlds, rather than an object generation task. We also successfully train timestep-augmented GFlowNets that fulfill the DAG-MDP criteria of GFlowNets.

**Policy Covers:**   Policy covers were introduced in (Du et al., 2019) to study exploration in episodic MDPs and serve as an inspiration for our work. A set of policies $\Pi$ is called a Policy-Cover of the state space $\mathcal{S}$ if for all states $s \in \mathcal{S}$, the probability of reaching that state within a given number of steps $h$ is high $\forall \pi \in \Pi$. Policy-covers have mainly been explored in theoretical RL problems, and have become a major tool for provable exploration in tabular MDPs (Lamb et al., 2022; Efroni et al., 2022; Mhammedi et al., 2023b). Policy covers have also been explored for provable exploration in the low-rank MDP setting, which is closely related to linear MDPs (Mhammedi et al., 2023a). In this work, we take inspiration from this line of research, and show how GFlowNets (Bengio et al., 2021) can be used to learn a set of diverse, and robust, policies forming a policy cover over the state and goal distributions.

**Exploration and Robustness:**   Exploration-exploitation tradeoff is one of the fundamental areas of reinforcement learning to ensure an effective learning that spans the whole space. One of the earliest, and simplest, methods to encourage exploration and stochasticity is randomized action selection in which a random action is selected throughout learning. A number of ways have been explored to select this random action selection (Sutton, 1995; Mozer & Bachrach, 1989; Burlington & Dudek, 2008). Another class of methods uses *optimism in the face of uncertainty* with an optimistic initialization, preferring actions with more uncertainty, or using disagreement between multiple models to estimate uncertainty Kaelbling et al., 1996; Thrun & Möller, 1991; Strehl & Littman, 2008; Jaksch et al., 2010; Szita & Lörincz, 2008; Rashid et al., 2020; Bellemare et al., 2016; Fu et al., 2017; Martin et al., 2017; Tang et al., 2016. *Bayesian methods* use a Bayesian framework to learn a posterior over actions or models to randomize actions and encourage exploration (Dearden et al., 1998; Wang et al., 2005; Poupart et al., 2006; Guez et al., 2012; Touati et al., 2018; Azizzadenesheli et al., 2018; Janz et al., 2018). This class of methods have a more structured way of exploration, but can be hard to scale. *Intrinsic motivation* based methods reward an agent to be curious to explore, independent of the extrinsic reward provided by the environment (Schmidhuber, 1991a;b; Pathak et al., 2017; Hazan et al., 2019; Amin et al., 2020; Scott & Markovitch, 1989; Hong et al., 2018; Little & Sommer, 2013; Shyam et al., 2018). The main methods used to derive an intrinsic reward fall into two categories. The first is to explore where there is high error in a prediction model, such as the environment dynamics. The intrinsic curiosity module Pathak et al. (2017) provides higher reward signal to states where there is high error in the dynamics and inverse dynamics model. The second is maximizing the novelty of visited states. Through state visitation counts, Bellemare et al. (2016) rewards the agent to visit novel states that have not previously been visited. Badia et al. (2020); Liu & Abbeel (2021) utilizes K-nearest embeddings of recently visited states to encourage exploration away from previously visited states. Recent state of the art methods have combined both categories (Zhang et al., 2021; Wan et al., 2023; Badia et al., 2020), resulting in efficient exploration of the state space. In contrast, to previous exploration work, our work does not require learning a separate model, to capture uncertainty or to calculate intrinsic rewards. Instead, we use a principled approach of training a diverse policy cover to efficiently explore the state space.

## 3    Background

To explain the details of our proposed method we will start with an outline of the planning space then a description of the GFlowNet framework.

### 3.1    Goal Conditioned Reinforcement Learning

In a Markov decision process (MDP) with states $s_t \in \mathcal{S}$, actions $a_t \in \mathcal{A}$, dynamics $P(s_{t+1}|s_t, a_t)$, rewards $r_t$, horizon $H$, and discount factor $\gamma$, reinforcement learning addresses optimizing a policy $\pi_\theta(a_t \mid s_t)$ to maximize expected return $\mathbb{E}[\sum_{t=0}^{H} \gamma^t r_t]$. We examine the partially observed MDP (POMDP) setting, where the state $s_t$ is not fully observed. In goal conditioned reinforcement learning, a goal $g_t \in \mathcal{G}$ is given at the start of each episode. The policy input is augmented $\pi_\theta(a_t \mid s_t, g_t)$ (Kaelbling, 1993), and can then be optimized over the distribution of goals. In this setting the reward function used for the agent is some distance $d(\cdot, \cdot)$ between the state and the goal, $r(s_t, g_t) = d(s_t, g_t)$, often an L2 distance or indicator function when the state $s_t$ is withing an $\epsilon$ distance to the goal $g_t$.

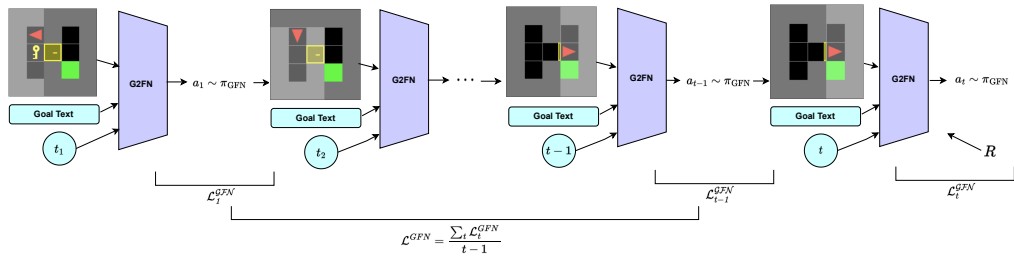

Figure 2: **G2FN:** Goal2FlowNets (labeled G2FN) set up through Goal-Conditioned GFlowNets to learn a set of policies that learn to reach the given goals.

Many different RL algorithms have been used to optimize this objective, but they do not explore the state space in a robust fashion.

### 3.2 GFLOWNETS

In this section, we discuss the elements of GFlowNets theory relevant to this work. For a more detailed treatment of GFlowNets see Bengio et al. (2023). Let $G = (\mathcal{S}, \mathcal{A})$ represent a Directed Acyclic Graph (DAG) such that $\mathcal{S}$ and $\mathcal{A}$ represent the set of nodes and the set of directed edges respectively. In a directed edge, $(u \rightarrow v) \in \mathcal{A}$, $u$ is called a *parent* of $v$ and $v$ is called a *child* of $u$. We define a unique *initial state* with no parents, $s_0 \in \mathcal{S}$, that also represents the root node, and define $\mathcal{X}$ as the set of *terminal* nodes with no children. A sequence of states $\tau = (s_m \rightarrow s_{m+1} \rightarrow \cdots \rightarrow s_n)$ is defined as a *trajectory* or an *action sequence* such that each $(s_i \rightarrow s_{i+1})$ represents an action. In a *complete* trajectory the initial state is $s_0$ and the terminal state is $s_n$ (typically in GFlowNets, terminal states correspond to some fully constructed object $x \in \mathcal{X}$). The set of complete trajectories is denoted by $\mathcal{T}$.

The *(forward) policy* of a GFlowNet represents a collection of distributions $P_F(-|s)$ over the children of every nonterminal state $s \in \mathcal{S}$. A forward policy, the equivalent of "the" policy $\pi$ in RL, determines a distribution over $\mathcal{T}$ as:

$$P_F(\tau = (s_0 \rightarrow \cdots \rightarrow s_n)) = \prod_{i=0}^{n-1} P_F(s_{i+1}|s_i). \tag{1}$$

In a GFlowNet, this forward policy can be used to sample states $x \in \mathcal{X}$ in an iterative manner: starting from the initial state $s_0$, each next action sampled from $P_F$ adds to the partial object constructed so far, until a terminating state is reached. The marginal likelihood of sampling $x \in \mathcal{X}$ is the sum of likelihoods of all complete trajectories that terminate at $x$. GFlowNets are flow-based networks that learn the forward policy $P_F$ such that the likelihood of sampling an object $x \in \mathcal{X}$ is proportional to the nonnegative reward $R(x)$ associated with $x$ such that the following is satisfied:

$$R(x) = Z \sum_{\tau=(s_0 \ldots s_n=x)} P_F(\tau) \quad \forall x \in \mathcal{X}. \tag{2}$$

If Eq. 2 is satisfied, then $Z = \sum_{x \in \mathcal{X}} R(x)$. As alluded to earlier, a number of objectives have been proposed to train GFlowNets. Due to its stability and sample efficiency, we use the SubTB($\lambda$) objective.

**Sub-Trajectory($\lambda$) balance (SubTB($\lambda$), Madan et al., 2023)** The Sub-Trajectory($\lambda$) balance objective was proposed to unify TB and DB, such that sub-trajectories of any length could be used to compute a loss and update a GFlowNet. GFlowNets using SubTB($\lambda$) are parameterized by three objects: a *forward policy*, $P_F(-|-; \theta)$ representing a distribution over children, a *state flow* function $F(s; \theta)$ representing the flow (unnormalized probabilities) through state $s \in \mathcal{S}$, and a *backward policy* model $P_B(-|-; \theta)$ representing a distribution over the parents of state $s \in \mathcal{S}$. Flows are matched with sub-trajectories $\tau = (s_m \rightarrow \cdots \rightarrow s_n)$ that can be of any length with

$$\mathcal{L}_{\text{SubTB}}(\tau; \lambda) = \frac{\sum_{m \leq i < j \leq n} \lambda^{j-i} \mathcal{L}_{\text{SubTB}}(\tau_{i:j})}{\sum_{m \leq i < j \leq n} \lambda^{j-i}}. \tag{3}$$

where

$$\mathcal{L}_{\text{SubTB}}(\tau_{m:n}) = \left(\log \frac{F(s_m;\theta) \prod_{i=m}^{n-1} P_F(s_{i+1}|s_i;\theta)}{F(s_n;\theta) \prod_{i=m}^{n-1} P_B(s_i|s_{i+1};\theta)}\right)^2.$$ (4)

where when $s_n = x$ is terminal, we take $F(x;\theta) = R(x)$. The hyperparameter $\lambda$ is used to take a weighted combination of all sub-trajectories, and happens to provide a smooth interpolation between prior objectives; $\lambda \rightarrow 0^+$ leads to the detailed balance objective (over all transitions in $\tau$), and $\lambda \rightarrow +\infty$ yields the trajectory balance objective (see Madan et al., 2023).

## 4 GOAL2FLOWNETS: GFLOWNETS FOR GOAL-CONDITIONED REINFORCEMENT LEARNING

GFlowNets have been used in a wide range of applications so far, and besides the work of Pan et al. (2023; 2022), have rarely been used to train reinforcement learning agents in a sparse-reward setting nor been leveraged for their inherent exploratory properties in RL. There have been recent studies exploring the potential of GFlowNets to discover diverse modes of the reward function, but they often overlook its capability to generate diverse *solutions*, trajectories, for a *single* given outcome. From an RL perspective, this is an *exploratory* behavior.

In this work, we use Goal-Conditioned GFlowNets (Fig. 2) such that the resulting policy can navigate its environment and complete the task, with an added benefit of GFlowNets: an inherently exploratory behavior that discovers many ways to complete a task in proportion to the "goodness" of that way. It should in fact be possible to push the model to discover *every* way, since by using the right $P_B$ we can induce maximum entropy over successful trajectories in GFlowNets (Zhang et al., 2022). To this end we would use a uniform $P_B$ instead of the learned $P_B$.

Formally, we define G2FNs using conditional $P_F$, $P_B$, and $F$, conditioned on a goal and the current timestep; we write $P_F(s'|s, g, t; \theta)$. This is achieved by concatenating the goal and the timestep to the state.

One of the assumptions of GFlowNets is that the underlying MDP is a DAG, meaning it should not have any cycles. In order to fulfill this condition, we augment the state observation with the current timestep. Another assumption of GFlowNets is that only terminal states contain reward; this is naturally compatible with goal-based setups, where episodes end upon reaching the goal. As in goal-conditioned RL, the goal description $g$ can be somewhat arbitrary as long as the reward $R(s, g)$ is consequent with it.

Another interesting side-effect of using GFlowNets to learn policies is that they, by design, attempt to put probability mass on every terminal state, even if just a little. This is yet another exploration mechanism, one which does not require any special heuristic unlike much of prior work in exploration in RL.

To summarize, our use of GFlowNets to learn policies naturally induces two exploration mechanisms without having to rely on any heuristic. Thanks to the off-policy learning capabilities of GFlowNets this exploration is sample efficient. In addition, although curiosity-driven exploration approaches can motivate the agent to explore diverse states, they generally do not induce the capability of producing diverse trajectories to these states, i.e., they opt for state-wise diversity instead of state-wise *and* trajectory-wise diversity. Future work should expand on this and include explicit exploration-based heuristics as well to target even more difficult RL problems.

## 5 EXPERIMENTS

In our experiments, we aim to answer the question: *how robust are Goal2FlowNets compared to other methods*. To help answer this question we perform three different kinds of analysis: (1) We study the ability of the Goal2FlowNet policy to be a good policy cover, i.e. to visit states and successful trajectories with a diverse distribution. (2) As a form of robustness, the sample efficiency is compared to prior methods, and (3) We evaluate the final policies on held-out environment configurations to understand the scope of distribution changes the method can overcome. For each of these experiments we use *success* as a measure of performance, where, we define success as the percentage of the agent's trajectories that reach the goal $g$, with an episode time limit of $H$.

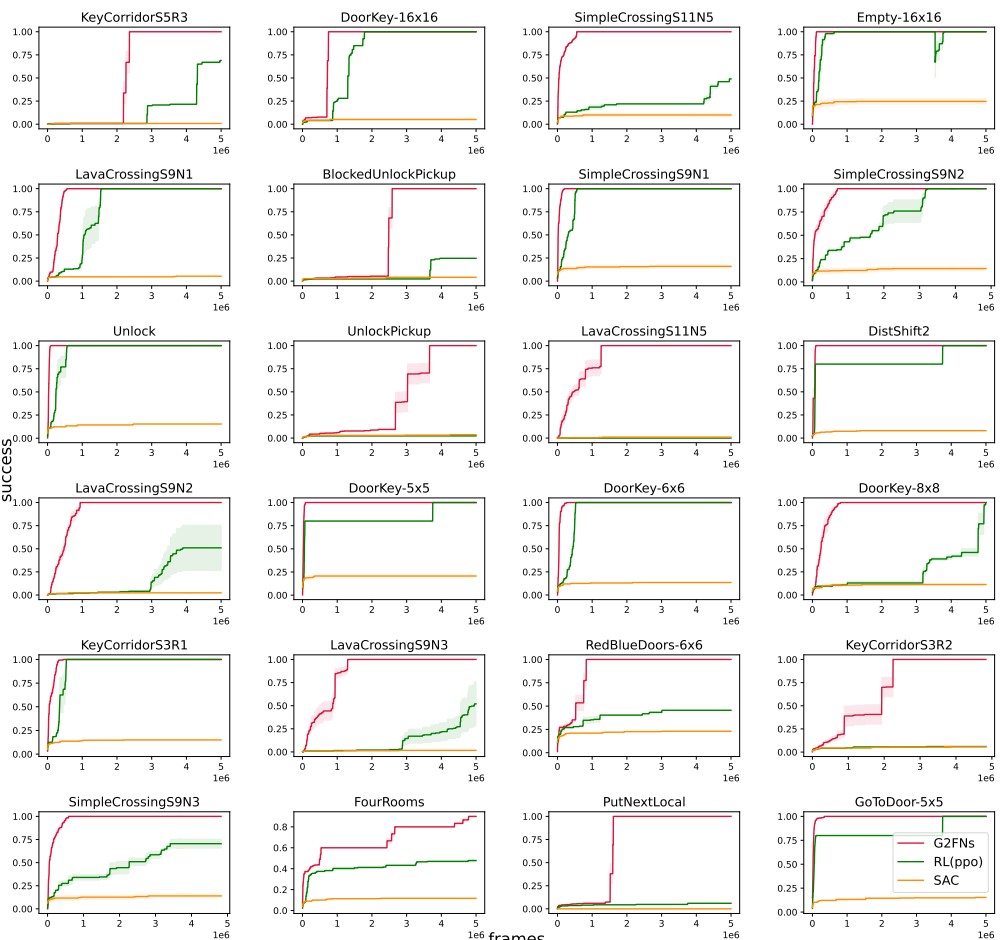

Figure 3: **Sample Efficiency:** Goal2FlowNets (labeled G2FNs) discover and learn to reach the specified goals with many fewer examples over a wide range of grid-world environments having partial egocentric observations and sparse rewards. The effect of sample efficiency becomes much more prominent as the complexity of the environments increases.

**Environments:** We train on a wide variety of tasks from the MiniGrid and BabyAI environments (Chevalier-Boisvert & Willems, 2018; Chevalier-Boisvert et al., 2018), and evaluate the performance of the learning agent every few frames. For these tasks, the state input observation consists of a partially observed egocentric view of the environment, and BabyAI gives goals in a textual format.

**Sample Efficiency** We find that Goal2FlowNet agent learns from a much smaller number of episodes / frames as compared to other methods, including those that use off-policy learning such as SAC (Haarnoja et al., 2018), as shown in Fig. 3. This shows that Goal2FlowNets can learn in an efficient manner on real reinforcement learning tasks.

**Changes to the Distribution of the Goal Position.** In order to understand the kinds of distribution changes that the learnt policies can handle, we train agents on one setting of partially ovserved environment in which the goal is always located at one fixed position. We test this agent on a similar environment, but with a different start and goal states. While other methods suffer when the agent's start and goal positions are changed, G2FNs can still stay robust, showing it is robust to changes in configuration, see Table 1.

**Zero-shot Generalization** In order to evaluate zero-shot generalization abilities, we train policies to convergence on one set of environments, and test them in a zero-shot manner, i.e. without any further fine-tuning, on a different set of environments such that the configuration and the difficulty level is

| Train Env | Changes | PPO | G2FN |
|-----------|---------|-----|------|
| Empty16x16 | within 25% | 0.82 ± 0.03 | 1.0 ± 0.0 |
| Empty16x16 | within 50% | 0.54 ± 0.05 | 0.80 ± 0.05 |
| Empty16x16 | within 75% | 0.45 ± 0.03 | 0.78 ± 0.03 |

Table 1: Changes to goal distributions within a certain % of the total diameter of the environment. We find that even if the underlying goal locations are changed on a given environment, G2FN performs better than PPO.

changed in the new environments. Results are shown in Table 2. In addition to the previous baselines, we additionally compare to DEIR (Wan et al., 2023), a recent exploration baseline. Unsurprisingly, the baselines collapse when the new environment has obstacles, due to their lack of trajectory diversity that is sampled during evaluation. A key takeaway is that although DEIR explores the state space very well, the resulting policy does not maintain diversity and also collapses.

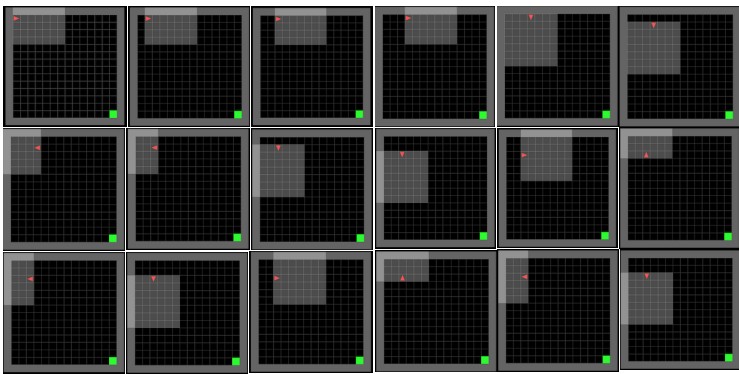

((a)) Zero-shot OOD (PPO): The agent stays within the training environment length & never reaches the goal

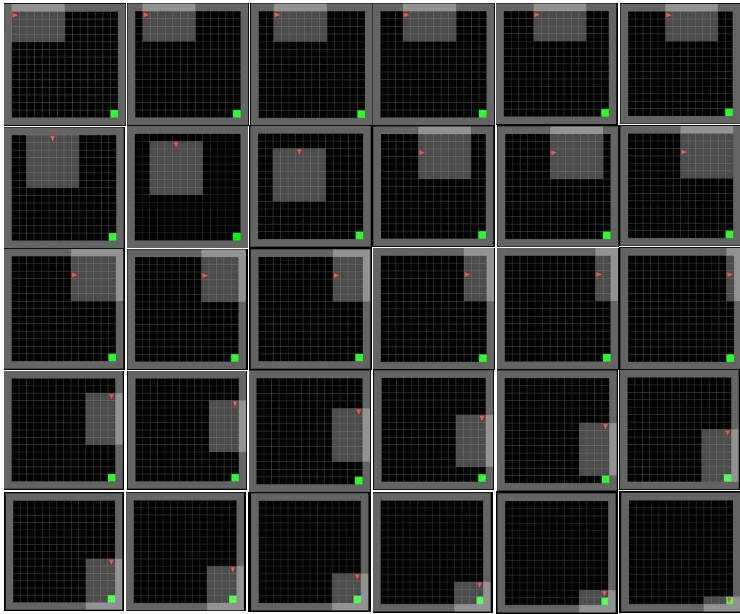

((b)) Zero-shot OOD for G2FN agent: The agent takes a much longer path to successfully reach the goal, highlighting its robustness and generalization properties with distribution changes.

Figure 4: Zero-shot OOD eval: G2FN shows robustness and generalization to distribution changes.

| Train env | Eval env | PPO | DEIR | G2FN |
|-----------|----------|-----|------|------|
| Empty-5x5 | Empty-8x8 | 0.85 ± 0.04 | 0.66 ± 0.04 | 1.0 ± 0.0 |
| Empty-5x5 | Empty-16x16 | 0 ± 0.0 | 0.88 ± 0.09 | 1.0 ± 0.0 |
| Empty-5x5 | CrossingS9N1 | .05 ± 0.01 | 0 ± 0.0 | 0.36 ± 0.02 |
| Empty-5x5 | CrossingS9N2 | 0.02 ± 0.01 | 0 ± 0.0 | 0.22 ± 0.03 |
| Empty-5x5 | CrossingS9N3 | 0.01 ± 0.01 | 0 ± 0.0 | 0.31 ± 0.03 |
| Empty-16x16 | CrossingS9N1 | 0.30 ± 0.02 | 0.30 ± 0.04 | 0.40 ± 0.01 |
| Empty-16x16 | CrossingS9N2 | 0.18 ± 0.04 | 0.16 ± 0.03 | 0.28 ± 0.03 |
| Empty-16x16 | CrossingS9N3 | 0.15 ± 0.04 | 0.09 ± 0.02 | 0.48 ± 0.01 |
| CrossingS9N1 | CrossingS9N3 | 0.89 ± 0.02 | 0.41 ± 0.06 | 0.94 ± 0.02 |
| DoorKey-5x5 | DoorKey-16x16 | 0.08 ± 0.01 | 0.0 ± 0.0 | 0.31 ± 0.03 |

Table 2: Zero-shot generalization: the agents are trained on Train env, and then tested zero-shot (without any further training) on a new set of considerably harder Eval environments. We find that Goal2FlowNets can learn to complete the task even when significant changes are made to the environment distribution. Reported are the success rate over 100 episodes, with the standard deviation over 3 seeds.

Goal2FlowNets did not collapse even when trained on the simplest version of the environments and tested on much larger and more difficult versions of these environments, showing its potential in terms of robustness and generalization to novel configurations and layouts. We also do a visualization for qualitative analysis of the performance of the paths taken by the learnt policies when evaluated in this zero-shot setting. Policies trained on the Empty-5x5 environment are visualized on the Empty-16x16 environment on which the PPO agent completely collapsed, see Fig. 4(a) and 4(b), and we can see that the PPO agent doesn't travel much beyond the environment length it has been trained on, while G2FN agent keeps moving in the right direction to find a path to the goal which is now located many steps away when compared to the training environment and successfully reaches the goal. This highlights the robust exploratory properties of G2FN that did not get stuck with a small subset of possible policies, but learns over the whole space of high-reward policies, making it easier to generalize or adapt when the environment changes.

**Learning-Free Few-Shot Generalization by Simulating Rollouts in a Perfect Model** Many reinforcement learning projects have leveraged a transition model (either perfect or learned) to improve the quality of the policy (Silver et al., 2017). This is especially true if the learned policy is diverse yet is only computationally tractable if the policy is effective. While G2FN achieves improved zero-shot generalization to novel environments, it is often still imperfect in some of the harder cases of distribution shift, such as when we train on an empty room and evaluate in an environment with patches of lava. We conjecture that G2FN has some coverage over policies which can handle the distribution shift, even if it puts some probability on a policy that fails. If this is indeed the case, then a model of the evaluation environment can be leveraged to achieve better performance in the new environment without having to re-train or explore in the new environment. To test this concretely we use the simulator itself as a perfect model and rollout an equal number of samples (such as $K = 100$) with each of the proposed methods, and then show that G2FN improves much more in this setting as compared to the baseline methods (Table 3).

**Policy Covers and G2FN** In simpler environments (such as traversing an empty gridworld) we can use combinatorics to determine the number of distinct optimal trajectories. We can also count the number of distinct trajectories which reach the goal but use more steps than necessary. By comparing this with the trajectories obtained from G2FN (and baseline methods) we can validate the degree to which we have obtained a policy cover. We train a G2FN on the Empty5x5 environment as it makes it easier to track the unique trajectories, and for this analysis, we remove the agent's direction from the observations to be able to directly move to the next cell in the grid. A trained G2FN agent traverses all the possible paths to reach the goal, reinforcing the benefits of a achieving wide coverage of the space through the learnt policy, Fig. 6.

| Train env | Eval env | Random Policy | PPO | G2FN |
|-----------|----------|---------------|-----|------|
| Empty-5x5 | Empty-8x8 | 0.30 ± 0.02 | 1.0 ± 0.0 | 1.0 ± 0.0 |
| Empty-5x5 | Empty-16x16 | 0.28 ± 0.01 | 0.0 ± 0.0 | 1.0 ± 0.0 |
| Empty-5x5 | CrossingS9N1 | 0.33 ± 0.01 | 0.34 ± 0.03 | 1.0 ± 0.0 |
| Empty-5x5 | CrossingS9N2 | 0.28 ± 0.01 | 0.10 ± 0.02 | 0.97 ± 0.02 |
| Empty-5x5 | CrossingS9N3 | 0.27 ± 0.03 | 0.10 ± 0.04 | 0.98 ± 0.02 |
| Empty-16x16 | CrossingS9N1 | 0.33 ± 0.03 | 0.95 ± 0.01 | 0.97 ± 0.01 |
| Empty-16x16 | CrossingS9N2 | 0.32 ± 0.01 | 0.80 ± 0.02 | 1.0 ± 0.0 |
| Empty-16x16 | CrossingS9N3 | 0.29 ± 0.01 | 0.70 ± 0.02 | 0.98 ± 0.01 |
| CrossingS9N1 | CrossingS9N3 | 0.32 ± 0.02 | 0.98 ± 0.01 | 1.0 ± 0.0 |
| DoorKey-5x5 | DoorKey-16x16 | 0.27 ± 0.02 | 0.30 ± 0.03 | 0.85 ± 0.03 |

Table 3: Learning-free few-shot generalization is nearly perfect when Goal2FlowNet is given the ability to do 10 simulated rollouts in the ground truth transition model. This suggests that even a few trajectories of gflownet achieve practically useful coverage of the state space.

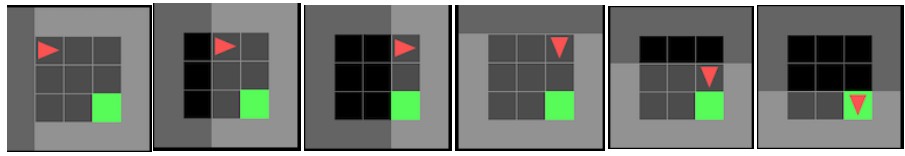

((a)) Trajectories taken by a converged PPO agent: the same path is almost always taken.

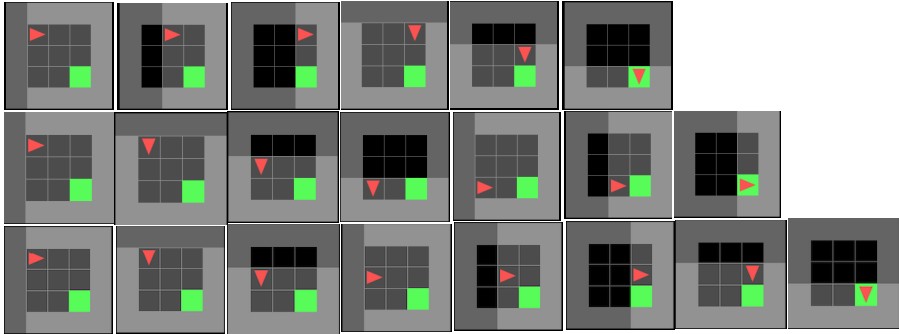

((b)) Trajectories taken by a converged G2FN agent: many paths can be taken to reach the goal.

Figure 5: Zero-shot OOD eval: G2FN show robustness and generalization to distribution changes.

**Visualizations of learnt policies and trajectories taken to reach the goal:** In order to qualitatively evaluating the trajectories taken by the trained models, we looked into the rollouts generated from a trained policy using RL(ppo) and G2FN. Whereas a converged PPO agent almost always follows the same path to reach the goal (keep going right and then move down, Fig. 5(a)), a converged G2FN agent traverses multiple paths to reach the goal. We show three such variations, in Fig. 5(b).

## 6 CONCLUSION

We have shown that goal-conditioned RL policies can struggle under mild adversity. In particular, we have shown that it is difficult to succeed with few samples, especially if the start states or goal states change, or if the environment changes. We have shown that a lack of diversity in the learned goal-conditioned policies is a major cause of these failures. We have shown that it is possible to address this limitation by learning diverse policies using a goal-conditioned variant of GFlowNet. This results in significantly improved sample efficiency as well as generalization to novel changes in the environment, such as introducing new obstacles. These results point towards a promising strategy for robust goal-conditioned reinforcement learning.

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

## 7 APPENDIX

### 7.1 MINIGRID AND BABYAI

BabyAI and MiniGrid (Chevalier-Boisvert et al., 2018; Chevalier-Boisvert & Willems, 2018) consists of a large variety of grid-based tasks in which a mission string defines the goal of the task. It was introduced to provide a framework for human-in-the-loop learning using a grounded natural language setting. Entities of different colors, such as the agent, balls, boxes, doors and key, are present within the environment and the goal or task is defined by a goal textual string. Agents can navigate with their environment and the objects present by picking, dropping and moving things around, and keys can be used to unlock doors in a mutli-door closer-door setting.

### 7.2 OBSERVATION SPACE:

The framework can provide a $7 \times 7$ egocentric partial view of the environment at every timestep. A fully observed input is also possible, but in this work, we mostly used the partial observed setting.

### 7.3 ACTION SPACE:

The framework provides a total of 7 actions, that can help an agent navigate and interact with its world. It is common to use only the relevant actions, but in our work, we allows all 7 actions.

### 7.4 SPARSE REWARDS:

The trajectories are rewarded only when the task is completed. If the task could not be completed, or if the time ran out before completing the task, no positive reward is given.

### 7.5 FIXED HORIZON:

Each environment has a maximum number of allowed step, also called horizon $H$, after which the trajectory is marked failure leading by zero reward.

### 7.6 EXPERIMENT SETUP

We use collect rollouts from 32 processes with 40 frames per rollout, and use Adam optimizer with learning rate $1e - 3$, $\alpha = 10^{-4}$, $\beta_1 = 0.9$, $\beta_2 = 0.999$ and $\epsilon = 10^{-5}$. For reinforcement learning experiments, we use a recurrent encoder that backpropagates through time at 20 steps. For PPO, we use 4 epoch updates. For G2FN, we do not use any recurrence. In order to prevent cycles, we add a timestep embedding to the input at each timestep, see Fig. 2. We maintain a replay buffer of successful trajectories that are used to make updates to GFlowNets in an off-policy manner. The returns by default are discounted, and we use this default version in this setup. Given max number of steps, or horizon, the final reward is defined as: $1 - 0.9n/n_{max}$, where $n$ is the number of steps taken and $n_{max}$ is the maximum steps allowed for that environment.

## 8 POLICY COVERS & G2FN AND STATE-SPACE COVERAGE

Since our work is inspired by Policy Covers, we visualize the space of trajectories covered by the a trained G2FN agent. For the Empt5x5 task in which it is feasible to track the paths taken, we find that the rolled out trajectories reach the goal by taking all possible optimal paths while covering the whole space, see Fig. 6.

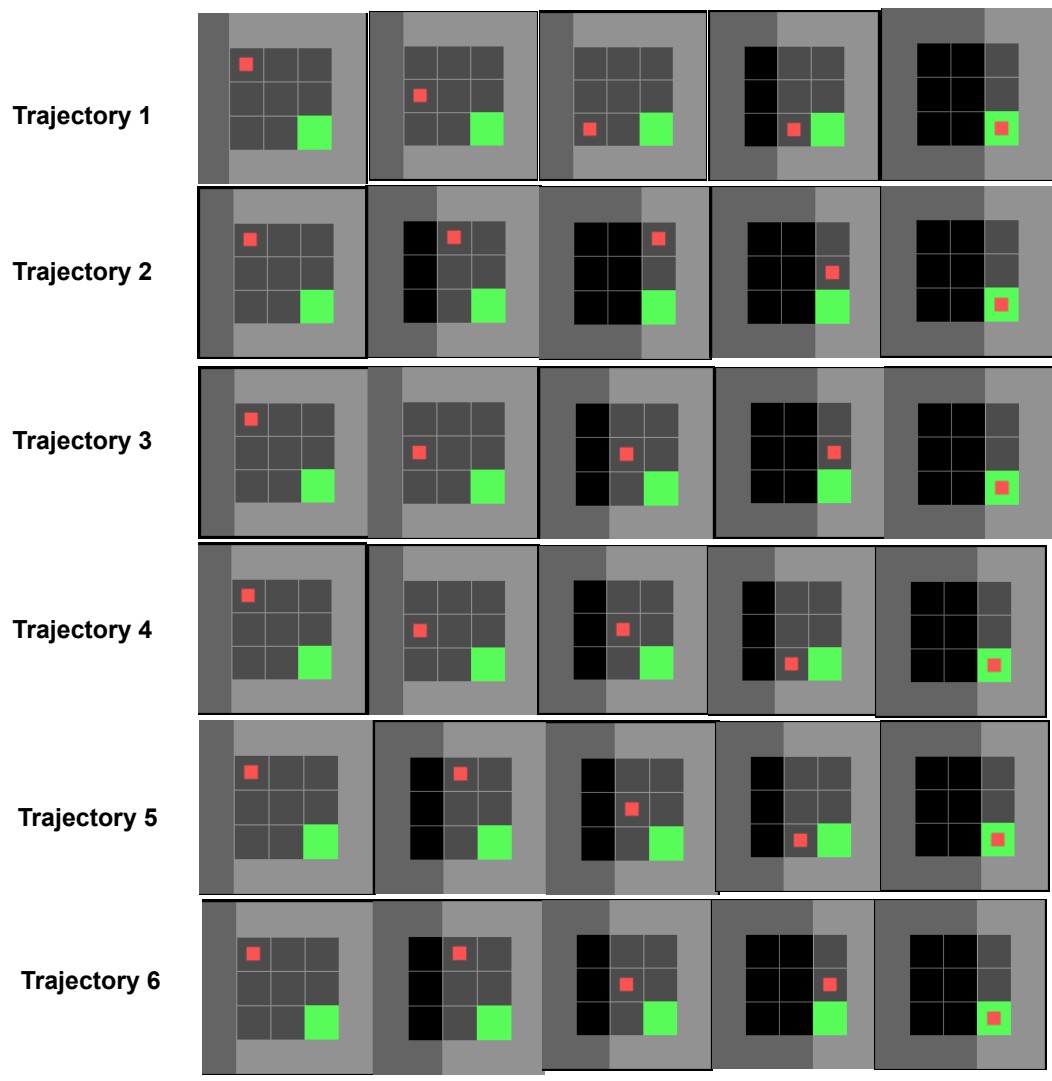

Figure 6: **Policy Covers & G2FN for state coverge:** a trained G2FN policy covers all possible paths to reach the goal state, reinforcing the inspiration of achieving a diverse policy cover through a G2FN policy.

