# OpenReview forum: "Goal2FlowNet: Learning Diverse Policy Covers using GFlowNets for Goal-Conditioned RL"
_ICLR.cc/2024/Conference — Submitted to ICLR 2024_

### Official Review · Reviewer_RmNo · 2023-10-31

**Soundness:** 2 fair
**Presentation:** 1 poor
**Contribution:** 2 fair
**Rating:** 3
**Confidence:** 4

**Summary:**

The paper presents a novel method called Goal2FlowNets for learning exploratory goal-conditioned policies in reinforcement learning settings. The proposed method employs Generative Flow Networks (GFlowNets) to learn robust goal-conditioned policies that can efficiently generalize to diverse policy covers and is capable of reaching goals through multiple optimal paths. Experiments conducted on various environments, such as MiniGrid and BabyAI, demonstrate that Goal2FlowNets significantly improves sample complexity, zero-shot generalization to new environmental changes, and coverage of the state and goal space.

**Strengths:**

- Originality: The proposed Goal2FlowNets method is an original approach to learning exploratory goal-conditioned policies, making use of the versatile Generative Flow Networks to enable better generalization, exploration, and adaptability in goal-conditioned reinforcement learning problems.
- Significance: The proposed method addresses a current limitation in goal-conditioned reinforcement learning methods by learning multiple nearly optimal paths to reach goals, resulting in better sample efficiency, zero-shot generalization, and overall improved agent performance in various environments.

**Weaknesses:**

- The paper has significant shortcomings in terms of problem modeling and training details. GFlowNet is adept at handling the generation of discrete objects (and later expanded to continuous scenarios) such as graphs and sets. However, the paper does not clearly specify the specific generated objects in the context of goal-conditioned RL using GFlowNet. Moreover, the paper lacks details on algorithm training and inference processes, such as pseudocode.
- The paper has gaps in the investigation of related work. The authors mention, "As far as we know, we are the first to use GFlowNets as policies in a typical RL setting". However, to my knowledge, there are at least two studies[1,2] that have already applied GFlowNet to RL tasks.
- Scalability: It is unclear how the proposed method performs in more complex or larger-scale environments. The presented experiments are limited to grid-world tasks, and the scalability of the method to more challenging environments remains unknown.
    - A discussion of the practicality of Goal2FlowNets in large-scale problems or more complex tasks would be valuable in assessing its applicability in broader contexts.
- Lack of ablation study: The paper does not provide an ablation study to understand the contribution of specific components of the proposed approach, such as the role of the timestep-augmentation in avoiding cycles in the underlying MDP.

---
[1] Li, Yinchuan, et al. "Cflownets: Continuous control with generative flow networks." ICLR 2023.

[2] Li, Wenhao, et al. "Diverse Policy Optimization for Structured Action Space." AAMAS 2023.

**Questions:**

- How do Goal2FlowNets perform on more complex environments or larger-scale tasks, and what challenges might arise when scaling up the method to these situations? There are existing works on applying GFlowNets to RL tasks [1,2], which have more complex experimental settings compared to this paper. It would be better for the authors to validate the scalability of the algorithm in non-grid-world scenarios.
- If the authors consider the trajectory from the source point to the target position as the generative object of GFlowNet, then I have doubts about the poor performance of the SAC algorithm. As stated by the authors of GFlowNet in Section 7.2 of this paper [3], GFlowNet has a relatively close relationship with Max-Ent RL. Specifically, Max-Ent RL tends to explore objects that require more time steps to construct. However, in the grid world, multiple trajectories reaching a specific target position have the same length, so in theory, Max-Ent RL should have similar performance to GFlowNet.

---
[3] Bengio, Yoshua, et al. "Gflownet foundations." Journal of Machine Learning Research 24.210 (2023): 1-55.

---

### Official Review · Reviewer_c4yU · 2023-11-02

**Soundness:** 3 good
**Presentation:** 1 poor
**Contribution:** 3 good
**Rating:** 3
**Confidence:** 4

**Summary:**

This paper proposed Goal2FlowNets, which uses Generative Flow Networks (GFlowNets) in order to learn exploratory goal-conditioned policies that are robust and can generalize better by learning multiple nearly optimal paths to reach the goals.

**Strengths:**

1. This paper is well-motivated and the algorithm seems simple, novel, and interesting (to my knowledge, I am not familiar enough with GFlowNet)

2. The experiment results seem good and the visualizations show good properties of this method.

**Weaknesses:**

1. The writing of this paper has to be fully revised. I do not get how the algorithm works, and the notation of this paper is confusing. For example, in sec 4, the authors keep saying that GFlowNets models $p(s'|s)$, however, the figure shows it is a policy that models $p(a|s)$.  Besides, what is $P_B$ and $F$? How does this algorithm work during training and inference?

2. The experiments only include the simple grid world. I wonder how this algorithm will work on complicated tasks, like image-based observations and continuous control problems.

**Questions:**

See above

---

### Official Review · Reviewer_vRuK · 2023-11-20

**Soundness:** 3 good
**Presentation:** 2 fair
**Contribution:** 2 fair
**Rating:** 3
**Confidence:** 5

**Summary:**

The core innovation of this paper is the proposal of Goal2FlowNets to form goal-conditioned policies. These policies learn multiple nearly optimal paths to reach goals. The authors highlight that their method allows for off-policy learning and implicitly captures epistemic uncertainty, which enhances sample efficiency and navigation of the state and goal space. Moreover, the policy offered by Goal2FlowNets is claimed to offer better generalization abilities in zero-shot and few-shot scenarios, particularly with sparse-reward and partial-observation environments (MiniGrid and BabyAI envs).

The baseline in this study compares against PPO and SAC. The main results show that G2FNs significantly improve sample efficiency compared to these standard RL methods in various grid-world environments. G2FNs also perform better in robustness and adapting to new environments without prior exposure.

**Strengths:**

1. Goal2FlowNets demonstrate better sample efficiency in learning to achieve goals, particularly in complex grid-world environments with partial observations and sparse rewards.

2. The proposed method is shown to be resilient to changes in the goal positions within environments, maintaining higher success rates compared to baselines like PPO in tests where the goal locations vary.

3. Goal2FlowNets display stronger zero-shot generalization capabilities, successfully handling tasks without additional training when confronted with significant alterations in environment distributions.

Diversity in exploration is one of the most fundamental problems in sequential decision making and we need more work in this area. In general I think this is an interesting line of exploration and warrants further investigation.

**Weaknesses:**

1. Unanswered questions and clarity: The paper is not very clearly written and there is a rich body of literature touching on similar topics. How is this related to distributional RL, RL as inference and using sequence/diffusion models to approximately solve the bellman's equation? There are so many unanswered questions left.

2. The paper may not discuss in detail how scalable the proposed method is to environments significantly larger or more complicated than the ones tested. It's unclear to what extent the improvements of Goal2FlowNets would generalize across a much broader range of domains outside the MiniGrid and BabyAI environments.

I think this approach needs to be scaled up to more complex environments with richer sensory information -- this will help answer some of the questions raised above

**Questions:**

1. Why were the experiments only restricted to environment with simpler observational spaces?

2. I think this paper lacks a proper theoretical foundation for decision making and this makes it very hard to contextualize it with respect to the existing RL literature. I would like to hear about steps towards this from the authors.

3. How do Goal2FlowNets balance the exploration-exploitation trade-off, especially in environments with high-dimensional state spaces or continuous action spaces? What about optimality?

4. What is the computational cost of employing Goal2FlowNets compared to traditional RL or deep RL methods?

5. Have the Goal2FlowNets been tested in environments with dense or shaped reward structures? How do they perform compared to environments with sparse rewards? "The trajectories are rewarded only when the task is completed. If the task could not be completed, or if the time ran out before completing the task, no positive reward is given." -> this seems like a big limitations?

---

### Meta-Review · Area_Chair_fjun · 2023-12-08

**Metareview:**

The paper presents Goal2FlowNets, a novel method for goal-conditioned policies. The key contribution highlighted is for improved sample efficiency, robustness, and generalization in environments like MiniGrid and BabyAI. However, the paper has significant limitations in its current form, including unclear writing, limited experimental scope, and insufficient theoretical grounding. There are also concerns about its scalability and novelty, as raised by the reviewers. The papers presentation needs more, has confusing details regarding the algorithm's functionality. Overall, while this paper shows promise, it falls short in clarity and comprehensiveness.

**Justification For Why Not Higher Score:**

N/A

**Justification For Why Not Lower Score:**

N/A

---

### Decision · Program_Chairs · 2024-01-16

Reject